# A targeted isotope dilution mass spectrometry assay for osteopontin quantification in plasma of metastatic breast cancer patients

Andrew Leslie[1], Evelyn Teh[1], Arik Druker[2,3], Devanand M. Pinto [1,4]*

1 Human Health Therapeutics Research Center, National Research Council, Halifax, Nova Scotia, United Kingdom, 2 Nova Scotia Health Authority, Division of Medical Oncology, Halifax, Nova Scotia, United Kingdom, 3 Faculty of Medicine, Dalhousie University, Halifax, Nova Scotia, United Kingdom, 4 Department of Chemistry, Dalhousie University, Halifax, Nova Scotia, United Kingdom

* dev.pinto@nrc.ca

## Abstract

Osteopontin (OPN) is a secreted glycophosphoprotein that derives its name from its high abundance in bone and secretion by osteoblasts. It is also secreted by a number of immune cells and, therefore, is present in human plasma at nanogram per millilitre levels where it affects cell adhesion and motility. OPN is involved in several normal physiological processes; however, OPN dyregulation leads to overexpression by tumor cells leading to immune evasion and increased metastasis. Plasma OPN is primarily measured by enzyme-linked immunosorbent assay (ELISA). However, due to the complexity of the various OPN isoforms, conflicting results have been obtained on the use of OPN as a biomarker even in the same disease condition. These discrepant results may result from the difficulty in comparing ELISA results obtained with different antibodies that target unique OPN epitopes. Mass spectrometry can be used to quantify proteins in plasma and, by targeting OPN regions that do not bear post-translational modifications, may provide more consistent quantification. However, the low (ng/mL) levels in plasma present a significant analytical challenge. In order to develop a sensitive assay for plasma OPN, we explored a single-step precipitation method using a recently developed spin-tube format. Quantification was performed using isotope-dilution mass spectrometry. The concentration detection limit of this assay was 39 ± 15 ng/mL. The assay was applied to the analysis of plasma OPN in metastatic breast cancer patients, where levels from 17 to 53 ng/mL were detected. The sensitivity of the method is higher than previously published methods and sufficient for OPN detection in large, high grade tumors but still requires improvement in sensitivity to be widely applicable.

## Introduction

In spite of advances in early detection and improved treatment, breast cancer continues to be the leading cause of cancer-related deaths in women [1]. Biomarkers can impact breast cancer

**Data Availability Statement:** The mass spectrometry data is available through PeptideAtlas accession number PASS04822.

**Funding:** DP and AD acknowledge grant support from the Atlantic Chapter of the Canadian Breast

Cancer Foundation. DP acknowledges support from the National Research Council Industrial Research Assistance Program. The funders had no role in study design, data collection and analysis, decision to publish, or preparation of the manuscript.

**Competing interests:** The authors have declared no competing interests exist.

treatment by helping identify aggressive breast cancer sub-types (prognostic biomarkers) or enabling treatment selection (predictive biomarkers). Once validated, biomarkers are often pursued as therapeutic targets; the human epidermal growth factor receptor 2 (HER2) is an excellent example. It was identified as a biomarker in 1987 [2] and Traztuzumab, a monoclonal antibody that specifically targets HER2, was approved in 1998. Traztuzumab and newer therapies targeting HER2, have dramatically improved breast cancer survival [3]. HER2 measurement is now used routinely to define prognosis and enable treatment selection. Osteopontin (OPN) is a glycophosphoprotein with a variety of functions in bone mineralization, cell adhesion, integrin signalling, T-cell suppression and cell motility. Dysregulation of OPN was implicated in the progression of several diseases, including breast cancer in 1993 [4] and, more recently, in several other solid tumors [5]. Unlike HER2, efforts to target OPN therapeutically have not progressed further than an unsuccessful phase I trial for inhibition of OPN in rheumatoid arthritis [6]. Therapeutic targeting of OPN is challenging due to the high circulating levels, isoform complexity and rapid protein turnover [7].

In spite of its small size (32 kDa), OPN possesses two integrin binding sites, one CD44 binding site, a thrombin cleavage site, seven glycosylation sites and up to twenty-nine phosphorylation sites [8]. In addition to post-translational modifications, alternative splicing gives rise to three variants, OPN-a, OPN-b and OPN-c [9]. This complexity presents challenges to both the design of both therapeutic and diagnostic assays for OPN. Nevertheless, using a novel monoclonal antibody against recombinant OPN, Singhal *et al* demonstrated that high plasma OPN levels were associated with deceased survival in metastatic breast patients [10]. Conversely, measurement of OPN levels in 253 breast cancer tumor samples concluded that OPN was not prognostic [11]. A more recent meta-analysis of high-quality studies with large patient cohorts and complete survival data, revealed that elevated tumor OPN is associated with a worse prognosis. Increased OPN was, on average, associated with a 3.7 times increase in the likelihood of death. However, the increased risk of death in individual studies ranged from 1 (no increase) to 21 times increased risk [12]. This large variability may be due to several factors, including variability in the methods used to quantify OPN.

The primary method for OPN quantification is the use of antibodies. However, the type of antibody and epitope targeted can vary significantly, perhaps leading to highly variable quantification. A recent study using in-house developed and commercial antibodies to measure plasma OPN levels in the same set of samples returned results that differed by over 300-fold, from 1.2 ng/mL to 396 ng/mL [13]. More recently, mass spectrometry (MS) based quantification of protein biomarkers has emerged as an alternative to antibody-based techniques. The relatively low levels OPN present in complex matrices complicates the analysis of OPN by MS. Nevertheless, techniques based on high-resolution mass spectrometry and tandem mass spectrometry are well suited to high-sensitivity quantification in complex samples. MS is an inherently multiplexed technique and can target multiple OPN epitopes in a single analysis. Sensitivity and overall assay performance can be further enhanced by combining MS with affinity purification and liquid chromatography (LC).

Quantification of plasma OPN by MS was recently demonstrated by Faria *et al* using a liquid chromatography—tandem MS (LC-MS/MS) [14] and by Zhou *et al* using time-of-flight tandem MS (TOF-MS/MS) [15]. Zhou *et al* use the high resolution of TOF and the specificity of MS/MS fragmentation to measure OPN in plasma. They also optimise OPN extraction from plasma using the affinity of OPN for DEAE-Cibricon Blue in order to detect OPN at 1 μg/mL levels. Affinity methods are also used by Faria *et al* but in this case OPN is extracted from plasma using a commercial anti-OPN antibody that is immobilised on streptavidin-coated plates. They take advantage of the multiplexed nature of LC-MS/MS to incorporate a stable-isotope labelled (SIL) peptide as an internal standard and achieve a lower limit of

quantification in plasma of 60.3 ng/mL for the signature peptide, GDSVVYGLR. In spite of these promising results, the measurement of multiple OPN epitopes at ng/ml levels in plasma by mass spectrometry has not been reported.

In this work, we develop a method for OPN quantification in plasma using LC-MS/MS that uses a direct, antibody-free approach for OPN extraction from plasma followed by quantification using isotope-dilution mass spectrometry. This simplified method makes use of protein precipitation, digestion and clean-up in a single device to minimise OPN losses. We evaluated several additional SIL peptides that correspond to OPN splice variants OPN-a, OPN-b and OPN-c since their roles in cancer are the most characterised amongst the several known OPN isoforms. {Briones-Orta, 2017 #29} The method is then applied to a series of plasma samples collected from metastatic breast cancer patients.

## Materials and methods

### Materials

LC-MS grade acetonitrile, HPLC grade isopropanol and lyophilized trypsin (Promega V5111) was purchased from Fischer Chemical. HPLC grade acetone was purchased from Caledon Laboratory Chemicals. ACS 98% grade formic acid was purchased from EMD Millipore. Dulbecco's phosphate-buffered saline (PBS) was purchased from Gibco. Multiple Affinity Removal Spin cartridges (MARS, HAS/IgG) were purchased from Agilent and ProTrap XG cartridges were purchased from Proteoform Inc. Ammonium bicarbonate was purchased from BioShop Canada Inc. A Milli-Q IQ 7000 system was used for water purification. Human pooled normal plasma purchased from Precision Biologic (ref#: CCN-10) was used for method development. Full length recombinant human osteopontin protein was purchased from Abcam (product number ab87460). Heavy-isotope labelled osteopontin tryptic peptides YPDAVATWLNPDPSQK, QNLLAPQNAVSSEETNDFK, AIPVAQDLNAPSDWDSR and ANDESNEHSDVIDSQELSK were purchased from ThermoScientific (AQUA Ultimate grade) consisting of 5 pmol/μL aliquots in 5% ACN in water. All C-terminal residues for the heavy-isotope peptides were labelled with $^{14}C$ and $^{15}N$.

### Patient samples

Peripheral blood (15 mL) was collected from 23 recurrent or newly diagnosed metastatic breast cancer patients. Blood was drawn and stored in EDTA vacutainer tubes. Patient samples were collected between September 2010 –September 2013. The eligibility criteria were age over 18 years, confirmed diagnosis of metastatic breast cancer and active follow up at the Nova Scotia Cancer Centre. The study protocol was approved by the Capital Health Research Ethics Board (Study Identifier: CDHA-RS/2009-088), Halifax, Nova Scotia and the National Research Council of Canada (2009–10).

### Methods

**LC-MS/MS.** Patient samples were analysed using an Agilent 1290 liquid chromatography (LC) system coupled to a QTRAP 5500 (AB Sciex Instruments) equipped with an ESI Turbo Spray ion source. The LC system consisted of a temperature controlled auto-sampler set to 5˚C (G4226A), a binary pump (G4220A), a column oven (G1316C) and was equipped with a 3 μm, 1 mm x 15 cm Thermo Acclaim PepMap 100 column (ThermoScientific). A Column-Shield 0.5 μm frit (Canadian Life Science, AS-850-1050) was used as a pre-column filter. All separations were performed using a flow rate of 50 μL/min with a column temperature of 40˚C and injection volume of 20 μL. Mobile phases used were water with 0.1% formic acid (A) and

acetonitrile with 0.1% formic acid (B). The following gradient was used for all separations: 5% B initially rising to 50% B at 20 minutes, 100% B held for 2 minutes then to 5% B in 2 minutes, a final 4 minute re-equilibration step at 5% B for a 30 minute total run time. Instrument settings for the QTRAP 5500 were as follows: source temperature 300˚C, ion spray voltage 5500, curtain gas 10, collision gas 5, entrance potential 10, collision cell exit potential 13. The nebulizer and auxiliary gases were set to 25.

**Protein precipitation.** Metastatic breast cancer patient samples were processed using the ProTrap XG filter cartridge system. Samples were stored at -80˚C and thawed on ice prior to analysis. 10 μL of thawed patient plasma was diluted with 90 μL of PBS and transferred to a plugged ProTrap XG filter cartridge. Immediately prior to the transfer of the diluted patient plasma, the filter of the ProTrap XG was wetted with 2 μL of isopropanol. The diluted plasma samples were then protein precipitated by the addition of 400 μL of acetone to the ProTrap XG filter cartridge. Samples were gently mixed by pipetting volumes up and down repeatedly and incubated at room temperature for 30 minutes. With the filter plug still attached the cartridge was placed in a 2 mL Eppendorf tube then centrifuged at 2500 $x\,g$ for 2 minutes. The plug was then removed and the now unplugged filter cartridge was centrifuged for 500 $x\,g$ for 3 minutes in a 2 mL Eppendorf tube with the flow-through being discarded. The protein pellet was washed by the addition of 400 μL of acetone to the filter cartridge and then centrifuged at 500 $x\,g$ for 2 minutes discarding the flow-through. The plug was replaced and the cartridge filter was wetted with 2 μL of isopropanol. The protein pellet was resolubilized directly in the filter cartridge by adding 400 μL of 20 mM ammonium bicarbonate with repeated gentle aspiration followed by sonication for 10 minutes.

**Protein digestion & internal standards.** A spiking solution of heavy-isotope OPN peptides was prepared by adding 10 μL of each peptide aliquot to 840 μL PBS in a 1.5 mL Eppendorf tube. A trypsin solution was prepared by adding 40 μL of 20 mM ammonium bicarbonate to a vial of 20 μg lyophilized trypsin (0.5 μg/μL). 15 μL of heavy-isotope spiking solution followed by 10 μL of trypsin was then added to each filter cartridge containing resolubilized protein. Samples were incubated overnight at 37˚C. Post incubation, samples were collected by placing the unplugged filter cartridge in a 2 mL Eppendorf tube and centrifuging at 2500 $x\,g$ for 5 minutes. The cartridge filter was then washed by adding 200 μL of 80% ACN with 0.1% formic acid and centrifuged at 2500 $x\,g$ for 5 minutes, collecting the flow-through in a 2 mL Eppendorf tube. For all samples, post-digest flow-throughs were combined, dried down and reconstituted in 5% ACN with 0.1% formic acid prior to LC MS analysis.

## Results and discussion

### MRM assay development

Peptides produced after tryptic digestion of recombinant OPN were analysed by LC-MS/MS in order to develop a sensitive and specific assay for plasma OPN. In order to provide coverage of the three well-described OPN isoforms, OPN-a, OPN-b and OPN-c, five peptides were selected from candidates predicted as highly observable in Peptide Atlas [16]. The optimal electrospray source settings, collision energy and declustering potential were determined empirically (Table 1 & S1 Fig). Analysis of 200 ng of digested OPN demonstrated that four of the selected peptides produced high signals of at least 1 x $10^6$ counts whereas one peptide, DSYETSQLDDQSAETHSHK, had a signal 1000 times lower and was not used further. Using the optimal conditions, the mass detection limit of the LC-MRM assay using synthetic peptides was 1.3, 1.5, 3.1 and 4.2 ng/ml for peptides QNLLAPQNAVSSEETNDFK, ANDESNEHSD-VIDSQELSK, AIPVAQDLNAPSDWDSR, and YPDAVATWLNPDPSQK respectively. This sensitivity is similar to the sensitivity of 2.5 ng/ml reported by Macur et al. [19]. Recombinant

**Table 1. MRM assay parameters.** 188 ng of OPN tryptic digest was injected and the collision energy (CE) and declustering potential (DP) were optimised. Each peptide was monitored using four transitions which are ranked in decreasing peak height. Low signal prevented determination of optimal settings for the peptide (DSY...SHK).

| Sequence | ID | Q1 m/z | Q3 m/z | Ion | Scaled Intensity | CE | DP |
|---|---|---|---|---|---|---|---|
| AIPVAQDLNAPSDWDSR | | 927.95 | 835.89 | $y_{15}$ | 1.0 | 32.5 | 80 |
| | | 927.95 | 862.37 | $y_7$ | 0.41 | 38.5 | 74 |
| | AIP | 927.95 | 933.41 | $y_8$ | 0.12 | 38.5 | 86 |
| | | 927.95 | 563.26 | $y_4$ | 0.07 | 47.5 | 86 |
| YPDAVATWLNPDPSQK | | 901.44 | 459.26 | $y_4$ | 1.0 | 43.2 | 89 |
| | YPD | 901.44 | 671.34 | $y_6$ | 0.29 | 40.2 | 89 |
| | | 901.44 | 546.26 | $b_5$ | 0.18 | 37.2 | 89 |
| | | 901.44 | 617.29 | $b_6$ | 0.07 | 34.2 | 89 |
| ANDESNEHSDVIDSQELSK | | 706.31 | 966.43 | $y_{17}$ | 0.80 | 19.9 | 65 |
| | AND | 706.31 | 806.39 | $y_7$ | 1.0 | 22.9 | 80 |
| | | 706.31 | 844.39 | $y_{15}$ | 0.70 | 22.9 | 95 |
| | | 706.31 | 679.33 | $y_{12}$ | 0.05 | 22.9 | 74 |
| QNLLAPQNAVSSEETNDFK | | 1053.011 | 356.1928 | $b_3$ | 0.90 | 53.6 | 92 |
| | QNL | 1053.011 | 243.1088 | $b_2$ | 1.0 | 56.6 | 89 |
| | | 1053.011 | 540.314 | $b_5$ | 0.53 | 41.6 | 83 |
| | | 1053.011 | 1056.448 | $y_9$ | 0.14 | 47.6 | 89 |
| DSYETSQLDDQSAETHSHK | | 1089.47 | 609.31 | $y_5$ | NA | NA | NA |
| | DSY | 1089.47 | 371.2 | $y_3$ | NA | NA | NA |
| | | 1089.47 | 596.22 | $b_5$ | NA | NA | NA |
| | | 1089.47 | 896.42 | $y_8$ | NA | NA | NA |

OPN protein was then spiked into normal serum in order to determine the sensitivity of the overall assay, including plasma processing, digestion and peptide isolation. A limit of detection limit (LOD) of 39 ±15 ng/mL and limit of quantitation (LOQ) of 97 ± 33 ng/mL was achieved for the AIPVAQDLNAPSDWDSR. Background values plus 3 times the standard deviation or five time the standard deviation were used to calculate the LOD and LOQ, respectively. For the QNLLAPQNAVSSEETNDFK peptide, the LOD was 128 ±45 ng/mL with a LOQ of 299 ± 92 ng/mL. The YPDAVATWLNPDPSQK peptide LOD was 209 ±67 ng/mL with a LOQ of 529 ± 131 ng/mL. The least sensitive peptide was ANDESNEHSDVIDSQELSK with a LOD of 636 ±261 ng/mL with a LOQ of 1688 ± 585 ng/mL. Transitions for the corresponding stable-isotope labelled peptides were then added to create the final assay.

Representative data from the analysis of OPN peptides in buffer at a concentration of 200 ng/mL is presented in Fig 1. As expected, this full length OPN-a isoform has all four peptides present. As isoform OPN-b lacks amino acids 59–72 (exon 5 deletion), no signal for peptide QNLLAPQNAVSSEETNDFK would be observed for OPN-b. In a similar fashion, OPN-c lacks amino acids 31–57 (exon 4 deletion) and no signal for peptide YPDAVATWLNPDPSQK would be observed for OPN-c. When summing the transitions, all four peptides were readily detected with a S/N ranging from 39 to 1500. The peptide AIPVAQDLNAPSDWDSR has two internal proline residues and, as expected, fragmentation to the corresponding $y_{15}$ and $y_7$ provided the highest S/N of all the transitions studied and was most suitable for low level OPN detection in plasma.

## Assay verification in plasma

The MRM assay was then used to measure plasma sample processed by protein precipitation using the Protrap XG device recently developed by Crowell *et al*. [17]. High-abundance plasma

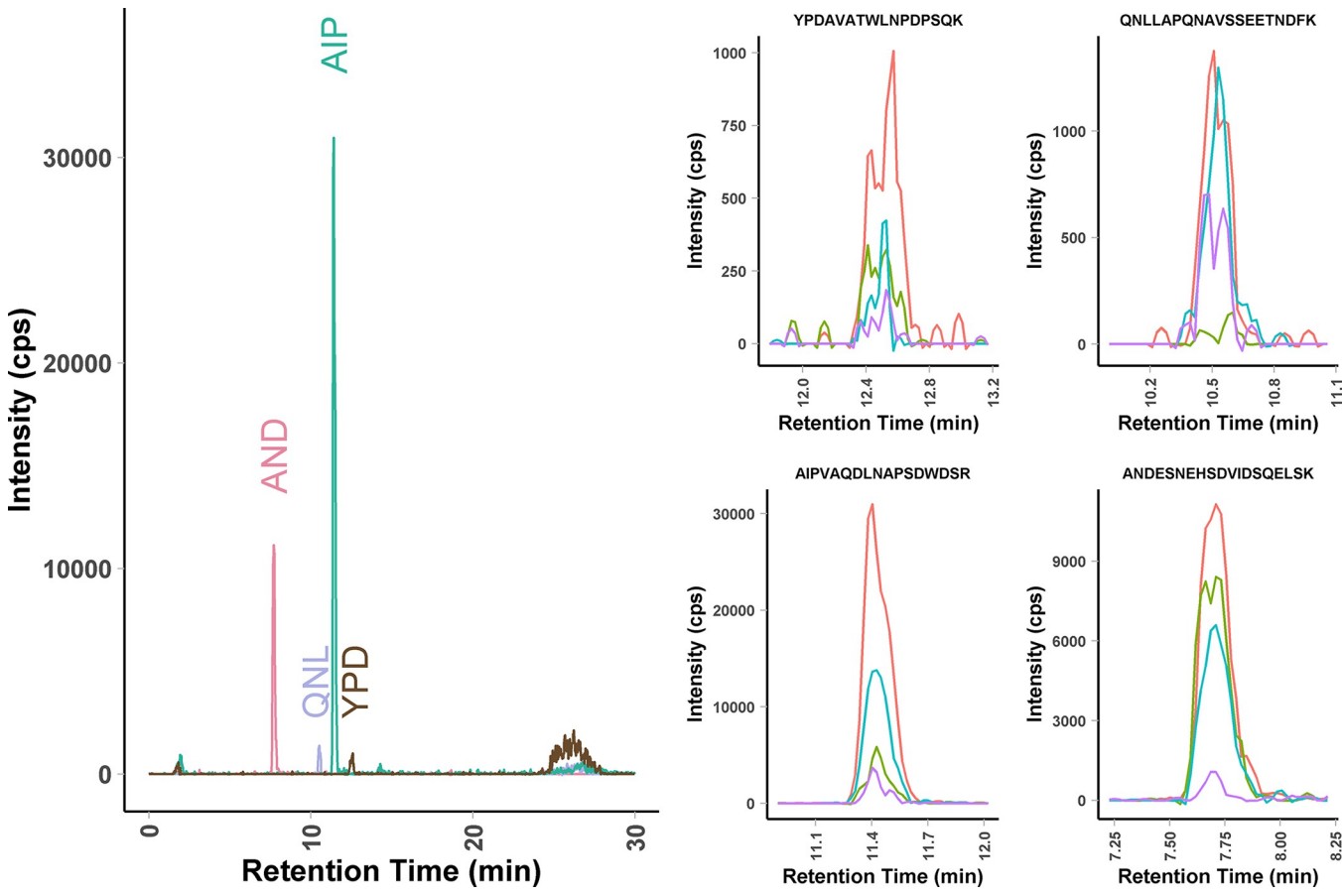

**Fig 1. Analysis of OPN standard by LC-MRM-MS.** The assay consists of four peptides with four fragments measured for each peptide for a total of 14 MRMs. In this example, a standard solution at 200 ng/ml in buffer was analysed. The use of multiple fragments provides high specificity.

proteins (HAPP) can often be a significant source of ion suppression and chemical noise in proteomic assays performed in plasma and serum. Therefore, we initially explored the utility of HAPP removal using the Agilent Multiple Affinity Removal Spin Cartridge System (MARS, HAS,IgG) to improve assay performance. Plasma from healthy controls was spiked with OPN at 1 000 ng/mL, which is approximately mid-point of the range (1–2,600 ng/mL) expected in metastatic breast cancer patients [18]. However, OPN losses were significant and precluded the use of this HAPP depletion approach for achieving low ng/mL sensitivity required for clinical sample analysis. OPN binding to reagents commonly used for depletion has been previously suggested [15]. Therefore, in order to minimise sample losses, a direct approach using the ProTrap XG device that used protein precipitation to remove salts, lipids and other interfering plasma components. This simplified approach allowed for the analysis of OPN without depletion or affinity steps and provided high signal to noise for the quantifier peptide (Fig 2).

This device uses filtration and an optional solid-phase extraction step (SPE) for processing of proteomic samples. Initial analysis using OPN protein in buffer processed using the manufacturer's protocol and comparison to solution digestion, revealed some loss of peptide signal. The most likely source of sample loss was adsorption to the filter material; therefore, after the trypsin digestion step, we added an additional 200 μL wash using 80% methanol/0.1% formic acid. When analysing recombinant OPN spiked into normal plasma, the additional elution step returned the recovery to 100% for all but one peptide, ANDESNEHSDVIDSQELSK,

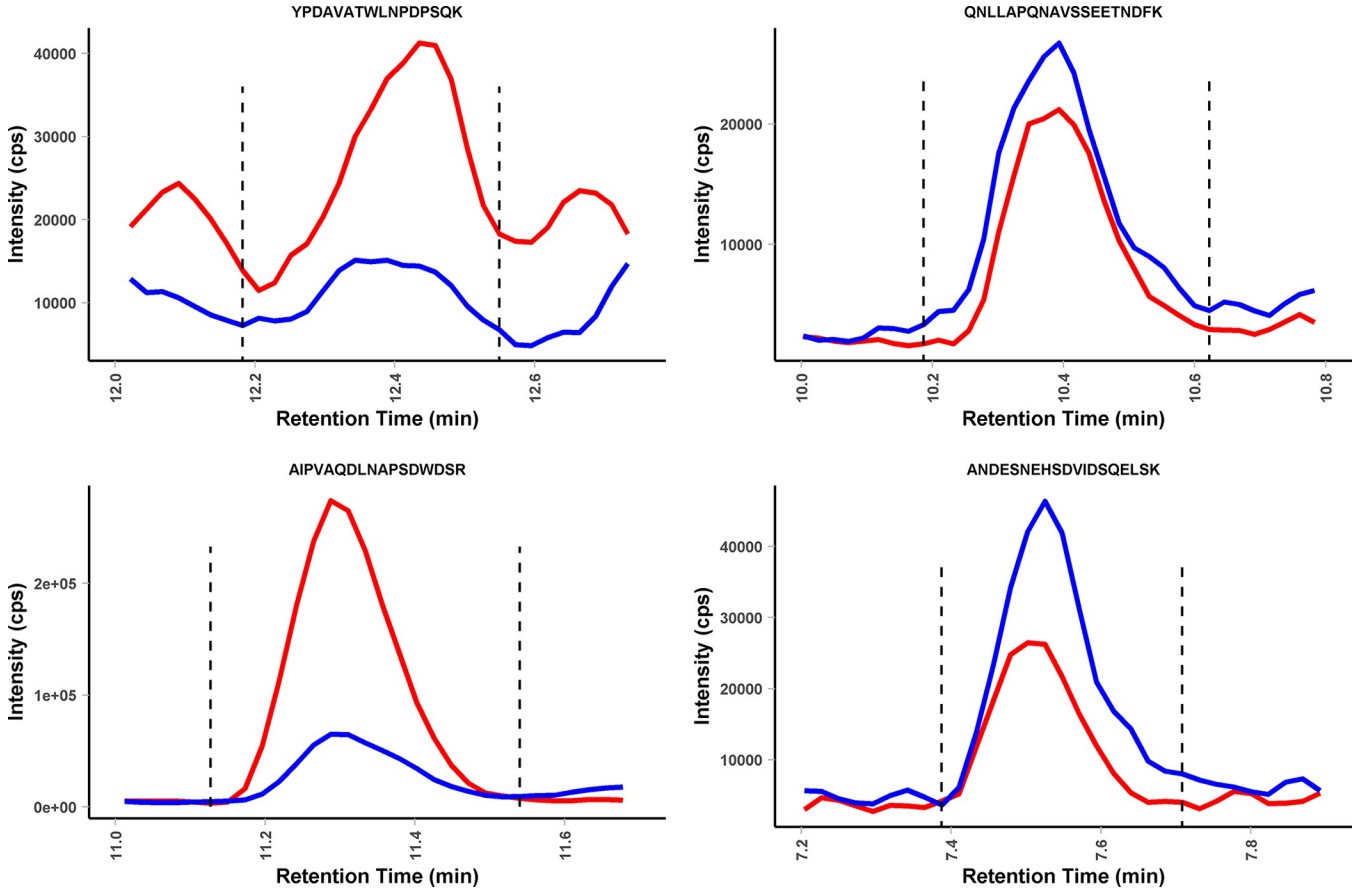

**Fig 2. Quantification of OPN spike in normal plasma.** OPN was spiked into control plasma at 400 ng/ml and processed using the optimised precipitation protocol. Quantification of the spiked protein (red traces) was performed using isotope dilution with a heavy labelled peptide (blue traces) for the four target peptides. The sum of four MRM transitions is shown.

which was the most hydrophilic of the four peptides (Fig 3). This additional elution step was used for all subsequent analysis.

## Measurement of OPN in plasma samples

As a proof on concept for the method developed, plasma from six triple negative breast cancer patients was used to demonstrate the applicability of the verified assay for endogenous OPN quantification. This sub-type was chosen based on the propensity to over-express OPN [18, 19]. All plasma samples were collected from patients with recurrent metastatic disease and have previously undergone multiple lines of systemic chemotherapy. Sample were labelled with anonymized identifiers (02, 03, 05, 06, 09 and 23) and the blood collection date. Patients 02 and 23 had plasma sampled at various time points. Patient 02, 09 and 23 had three, two and one metastatic sites, respectively when blood sample was first collected. Imaging studies for Patients 03, 05, 06 were not available to determine the number of metastatic sites. The OPN levels detectable from plasma ranged from 17 ± 1 ng/mL to 53 ± 4 ng/mL, with the lowest level found in the pooled normal controls and up to a 3-fold increase in the triple negative breast cancer samples (Fig 4). These values are between the assay LOD and LOQ, therefore, carry a large uncertainty and point to the need for further improvement in assay sensitivity.

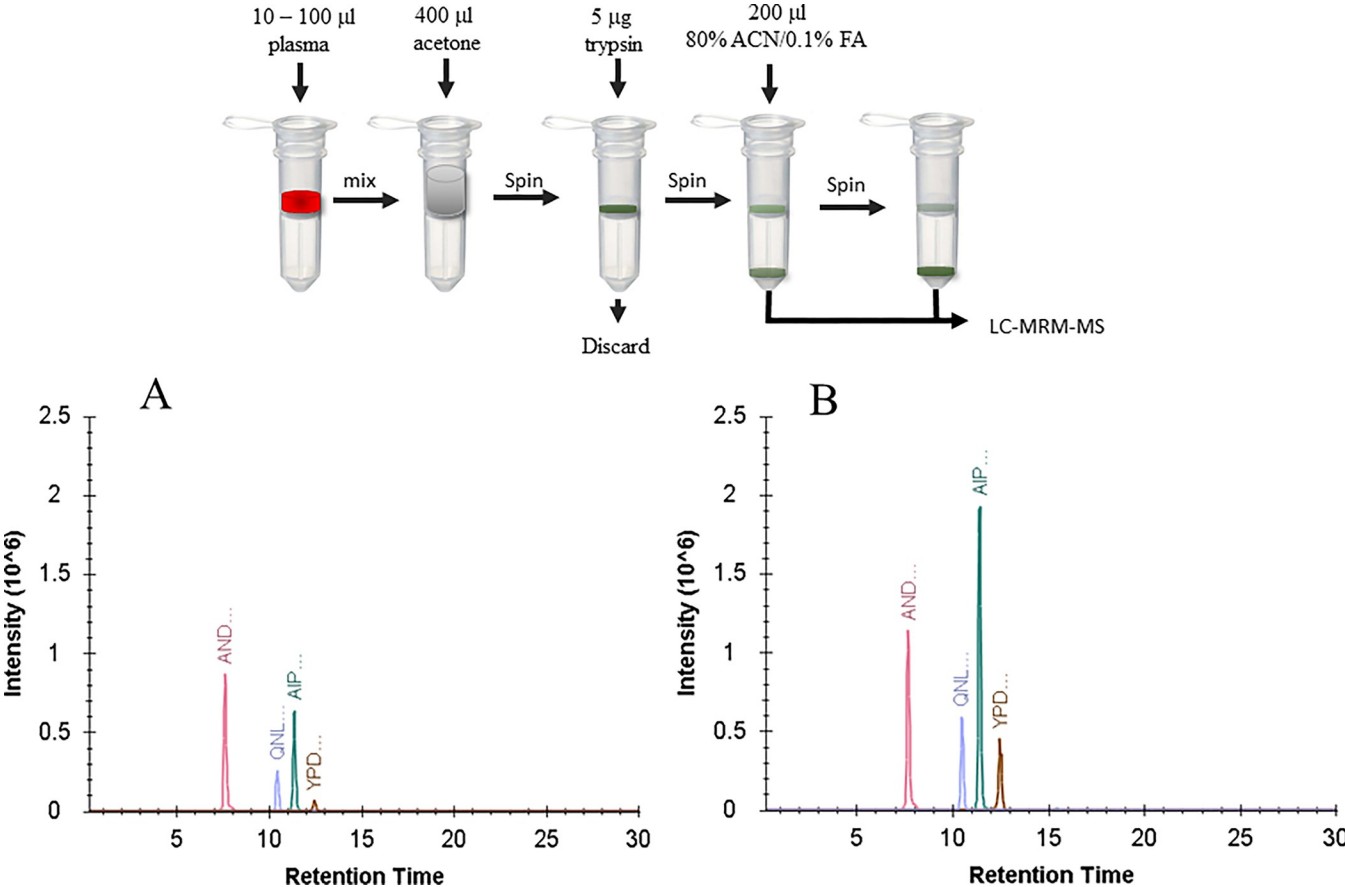

**Fig 3. Workflow and optimization of peptide recovery using addition extraction step.** Diagrammatic depiction of OPN extraction procedure using single-tube extraction, digestion process (top panel). Recovery was initially low (panel A) when the manufacturer's protocol was used but increased significantly when an additional extraction step was introduced (panel B).

Plasma OPN levels in breast cancer have been shown to vary quite extensively. In an ELISA based quantification, Bramwell et al reported OPN levels in metastatic breast cancer ranging from 1 ng/mL to 2648 ng/mL [18]. In another study, Plumer et al used in-house developed antibodies to measure concentrations of plasma OPN and found elevated levels in their metastatic breast cancer cohort (4.76 ng/mL) compared to healthy controls (1.2 ng/mL). On the other hand, OPN measurements by LC-MS/MS coupled to immunoaffinity by Faria et al ranged from 85–637 ng/mL in their breast cancer patient samples. Such variations are likely a result of the different assay formats, antibodies and peptides used to measure OPN concentration. It could be speculated that the low levels of OPN (similar to the pooled normal controls) in some of the cancer patients could be a result of systemic chemotherapy treatment. Slightly lower levels of mean plasma OPN were observed in metastatic breast cancer patients receiving systemic chemotherapy [18]. Serum OPN levels have also been found to decrease after chemotherapy in small cell lung cancer [20].

The correlation between plasma OPN levels and disease activity was examined in Patient 02 that had serial sampling between October, 2010 and April, 2011 (Fig 5). Where possible, plasma OPN was compared to tumor size assessed from CT imaging; however, since CT imaging is only performed every 2–4 months and blood draws are performed more frequently, this comparison is not possible at all time points. Plasma OPN levels increased concomitantly with

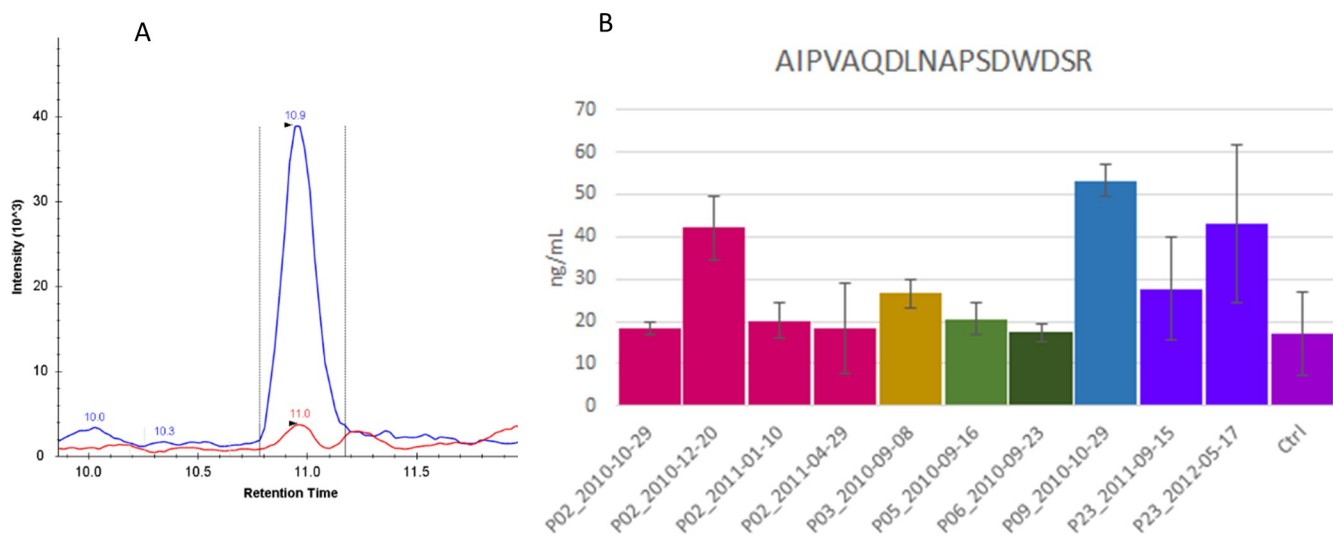

**Fig 4. Quantification of OPN in plasma of metastatic breast cancer patients.** Plasma OPN measurements based on the most sensitive peptide AIPVAQDLNAPSDWDSR from plasma of metastatic breast cancer patients. A) Extracted ion chromatogram (XIC) of a processed plasma sample (10 μL) following the established workflow. Plasma OPN concentration (red trace) was calculated using the ratio of the endogenous peak area over the heavy AQUA chain (blue trace). B) Bar plot representation of the OPN concentrations calculated from the plasma samples (mean ± SD). The control sample is from pooled normal human plasma.

disease progression as observed by CT imaging; however, plasma OPN levels were lower in subsequent sampling despite the lack of change in disease activity based on CT imaging. The specific disease progression observed for patient 02 was progression from primary to metastatic disease with new lesions being identified in the bone, which may have been facilitated by the increased OPN levels {Zuo, 2021 #30}. However, due to disease progression, the patient was also administered systemic chemotherapy, which has been shown to decrease OPN levels.

OPN has many diverse biological roles associated with aggressive cell behaviour, tumor progression and metastasis (Bramwell et al, 2006). High levels of OPN are associated with an increased tumor burden and decreased survival (Wang et al, 2020). The differential expression of OPN in breast cancer is still poorly understood and confounding variables of the host response to systemic chemotherapy may play a role in levels of plasma OPN and, therefore, its clinical utility is monitoring disease progression.

## Conclusion

The method developed is relatively straightforward; a batch of 8–16 plasma samples can be prepared and analysed in 24 hours, with approximately 3 hours of hands-on time. The method requires the use of stable-isotope labelled peptides, which are readily available and inexpensive, but does not require the use of antibodies, which can be difficult to produce reproducibly. The sensitivity of the method, with a detection limit of 39 ng/mL OPN in plasma, can detect OPN levels in metastatic breast cancer patients with high tumor burden but is insufficient for detection of OPN in early-stage breast cancer patients. The sensitivity is likely sufficient for the analysis of tumor biopsy material since OPN levels are often highly elevated at the tumor site. However, further increases in sensitivity are needed enable measurement of OPN at single ng/ml levels required to enable OPN isoform analysis using a liquid biopsy. This is advantageous as the is less invasive and can be performed a multiple time points. In addition, further increases in assay sensitivity would enable full coverage of the various isoforms of OPN in plasma in order to better understand the role of OPN isoforms in cancer progression.

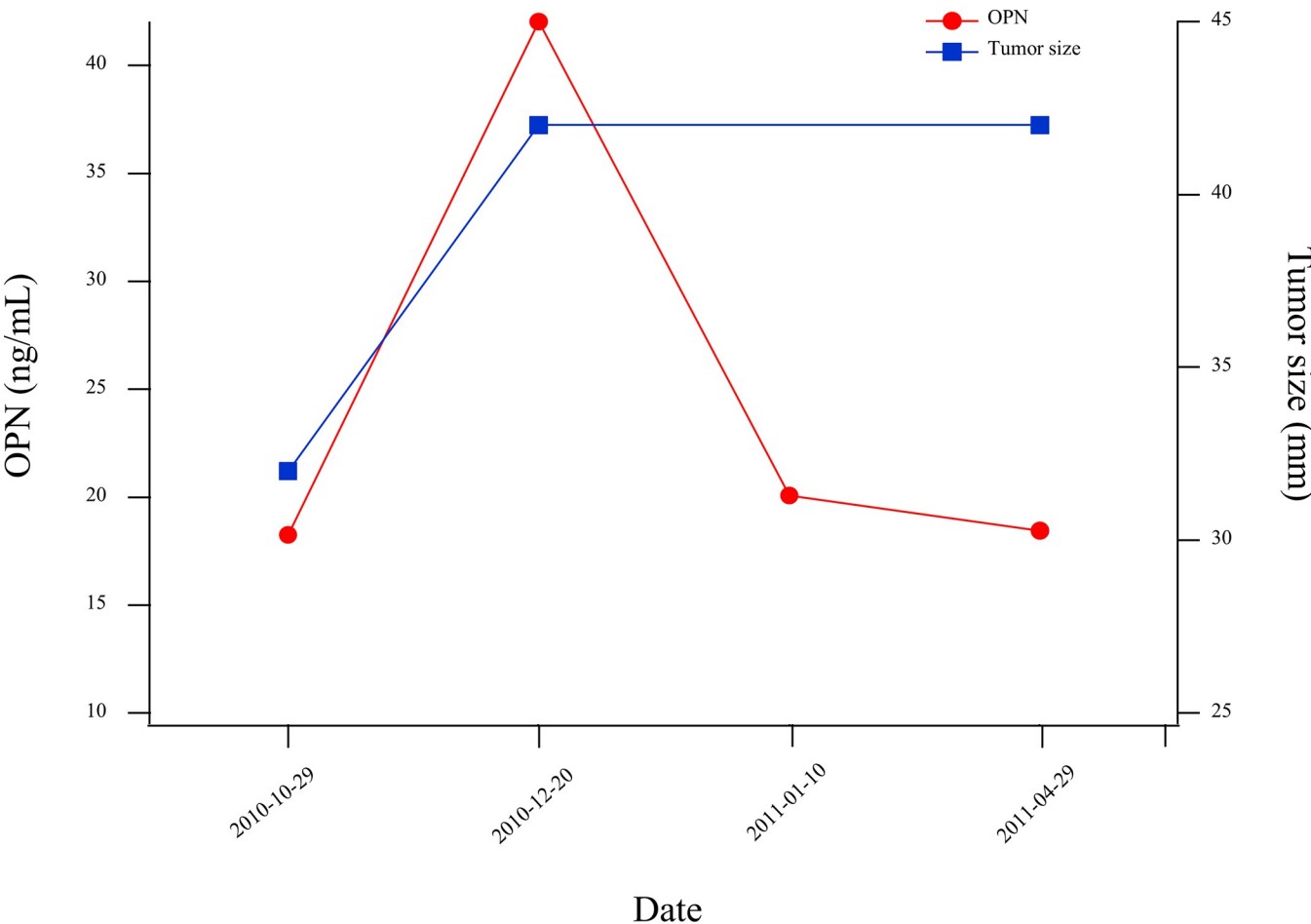

**Fig 5. Serial OPN measurements and tumor size.** Serial tumor size measurement by CT reveals rapid tumor growth accompanied by spike in plasma OPN concentration in Patient 02 suggesting change in OPN concentrations may be used to monitor disease progression.

## Supporting information

**S1 Fig. Collision energy optimization curves.** Optimization of collision energy for five target OPN tryptic peptides. A 200 ng of tryptic digest of OPN protein was injected for parameter optimization.
(TIF)

## Author Contributions

**Conceptualization:** Devanand M. Pinto.

**Funding acquisition:** Arik Druker, Devanand M. Pinto.

**Investigation:** Andrew Leslie, Evelyn Teh, Devanand M. Pinto.

**Methodology:** Andrew Leslie.

**Project administration:** Evelyn Teh, Arik Druker.

**Resources:** Evelyn Teh.

**Supervision:** Arik Druker.

**Writing – original draft:** Andrew Leslie, Devanand M. Pinto.

**Writing – review & editing:** Andrew Leslie, Evelyn Teh, Arik Druker, Devanand M. Pinto.

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
