## [Decision Letter · Decision Letter 0]

15 Feb 2023

PONE-D-23-02025A targeted isotope dilution mass spectrometry assay for Osteopontin quantification in plasma of metastatic breast cancer patientsPLOS ONE

Dear Dr. Pinto,

Thank you for submitting your manuscript to PLOS ONE. After careful consideration, we feel that it has merit but does not fully meet PLOS ONE’s publication criteria as it currently stands. Therefore, we invite you to submit a revised version of the manuscript that addresses the points raised during the review process.

ACADEMIC EDITOR: As appended below, the reviewers have raised major concerns/critiques and suggested further justification/work to consolidate the findings. Do go through the comments and amend the MS accordingly 

We look forward to receiving your revised manuscript.

Kind regards,

A. M. Abd El-Aty

Academic Editor

PLOS ONE

Reviewers' comments:

Reviewer's Responses to Questions

**Comments to the Author**

1. Is the manuscript technically sound, and do the data support the conclusions?

Reviewer #1: Yes

Reviewer #2: Yes

Reviewer #3: Yes

2. Has the statistical analysis been performed appropriately and rigorously? 

Reviewer #1: N/A

Reviewer #2: I Don't Know

Reviewer #3: No

3. Have the authors made all data underlying the findings in their manuscript fully available?

Reviewer #1: Yes

Reviewer #2: No

Reviewer #3: No

4. Is the manuscript presented in an intelligible fashion and written in standard English?

Reviewer #1: Yes

Reviewer #2: Yes

Reviewer #3: Yes

5. Review Comments to the Author

Reviewer #1: In the current manuscript, the authors have conducted a pilot study using a targeted isotope dilution mass spectrometry assay and quantified Osteopontin in plasma of six metastatic triple negative breast cancer patients.

The article is well structured into sections and subsections. The article is eloquent, and English is professional. It is within the scope of the journal.

However, there are some comments that need to be addressed to improve the article. The detailed comments are below:

1) Abstract, line 35: OPN is a multifunctional protein and is important for normal physiological processes. The dysregulation in its levels (overexpression) promotes tumor progression. The sentence needs to be rephrased to improve clarity.

2) Introduction: Is there a specific proteoform of OPN that is associated with the disease (cancer in this case)? For instance, elevated levels of glycosylated or phosphorylated forms? Not all proteoforms of OPN may be linked to cancer.

3) Introduction, line 89: Authors have described studies that have attempted OPN quantification. It would be relevant to describe Macur et al (2019) (Reference 19), where the analytical workflow included the enrichment of peptides and LC-MRM-MS analysis with stable isotope standards (SIS) standards that enabled the OPN quantification upto 2.5 ng/mL. Authors can highlight the merits of the workflow used in the current study, however the detection limit of this assay is 39 ng/mL.

4) Lines 217-224: For the pilot study, authors have used plasma from six triple negative breast cancer patients. However in line 221 and 222 authors mention patient 23. The sentence needs clarity. For instance, it can be mentioned as six patients namely or labelled 02, 03…and 23.

5) Lines 241-243: It is mentioned that plasma OPN levels increased with disease progression, then why plasma levels were lower in subsequent sampling?

6) Figure 1 Legend, line 264: The figure panels can be labeled and legend can be improved to aid the readers. Denote the short name used for the peptides either in Table1 or in the legend.

7) Figures: All figures need improvement. The image resolutions are very low, and the axis labels are not legible in some cases. Proper scaling is required. Figure 2 – The axis label seems half cropped. Figure 4 – nothing is legible.

Reviewer #2: Osteopontin was proposed as a potential biomarker in different diseases including breast cancer. Yet, to verify its biomarker potential in certain disease osteopontin needs to be verified in larger groups of clinical samples representative for a certain disease or its state. This in turn requires an efficient, high throughput and reliable methodology for sample processing and measurement. This goal was achieved by the authors of the manuscript, who developed a LC-MRM-MS-based assay with SIL standards for quantification of OPN in plasma with an efficient sample preparaction method that included protein precipitation, digestion and clean-up on a ProTrap XG filter cartridge. They verfied the assay on a set of breast cancer patients' plasma. In my opinion the manuscript is good, although needs some minor corrections/comments to be published in PLOS ONE:

(1) Please provide the ProteomeXchange number for your data-now it is missing.

(2) As I understood, the assay is intended to detect all OPN isoforms present in the sample, without distinguishing between them. Is any of the peptides unique for a particular isoform? Moreover, the concentration calculation is based on the signal detected for all the investigated peptides. Then, please comment on the impact of presence/absence of certain isoform on the final quantification results. The authors also mention post-translational modifications of OPN. PTMs were also detected on OPN peptides that they choose for quantification (please see eg. UniProtKB), which will make such modified peptide not detected by the developed LC-MRM-MS assay. Please comment on impact of presence/absence of the PTMs on the final quantification results and possible method limitations.

(3) Are LODs, LOQs and concentrations in patients's samples presented as mean +/- SD? Please specify and comment on the obtained results in the manuscript (line 180-187).

(4) Were calibration curves prepared for the investigated peptides or was it a single point calibration? Please specify clearly in the manuscript. If available, please add information about linear range of the assay.

(5) What software was used for data acquisition, extraction and processing? What parameters of the software were applied for data processing?

(6)I think it should be 'transtuzumab' instead of 'tranztuzumab' (line 55-56).

(7) It would be informative to give information about patients, where the cancer type and stage would be assesed by one of the clinically applied scales. Please include this information if available.

Reviewer #3: In their manuscript “A targeted isotope dilution mass spectrometry assay for Osteopontin quantification in plasma of metastatic breast cancer patients”, Andrew Leslie et al. developed 4 SID-SRM LC-MS assays for the quantification of the most relevant Osteopontin proteoforms. They also demonstrate the ability to quantify OPN in plasma samples using a slightly optimized sample preparation workflow that does not include any OPN enrichment step, which would be required to achieve sufficient sensitivities. As such, the targeted MS assays are sound, but lack sensitivity in real plasma samples. Nonetheless, the assays are a good starting point to use targeted MS for OPN quantification and are certainly interesting for researchers working in the field. I summarized my comments below that need to be addressed before publication.

Comments:

Line 212: The authors claim to recover 100% of the 4 selected peptides with their optimized sample preparation workflow. It is not clear from Figure two how this was determined. This is not too easy. Ideally, a reference protein would be spiked before the sample preparation in know amounts and is then determined by using heavy reference peptides spiked directly before LC-MS analysis. The ratio will then reflex the losses taking place during sample preparation. The authors should explain in detail how the 100% were determined or tune down this sentence saying that recovery of hydrophobic peptides was considerably improved by the additional elution step.

Line 225: The OPN concentrations found here ranged between 17 and 53 ng/ml. However, above, the authors describe the LOD/LOQs of the single peptide assays developed to be much higher. Only LOD of one peptide was at 39 ng/ml, LOQ at 97 ng/ml. Does this mean that the OPN concentrations determined are all below LOQ? How useful are the assays actually with LOQs being much above the actual OPN concentrations found in most plasma samples? This and further improvement possibilities to boost sensitivity should be discussed!

The entire data analysis part including a statical analysis is missing in the manuscript and should be added.

Figure 2: It would be helpful to add the time span between the spins into the schematic overview to better grasp the workflow. In particular, when adding trypsin, some incubation time (overnight) is required before spinning down the peptides.

Figure 5: Why was there no Tumor size measurement at 2011-01-10?

The raw MS data needs to be public available. Please upload to a public repository, like PANORAMA.

6. PLOS authors have the option to publish the peer review history of their article (what does this mean?). If published, this will include your full peer review and any attached files.

Reviewer #1: **Yes: **Ankita Punetha

Reviewer #2: No

Reviewer #3: No

---

## [Author Response · Author response to Decision Letter 0]

24 Apr 2023

February 22, 2023

A.M. Abd el-Aty

Academic Editor

PLOS ONE

Re: Reviewer comments for “A targeted isotope dilution mass spectrometry assay for Osteopontin quantification in plasma of metastatic breast cancer patients”

We would like to thank you and the reviewers for their careful consideration of our manuscript. Please find below our responses to their comments and concerns.

Reviewers' comments:

Reviewer's Responses to Questions

Comments to the Author

Reviewer #1: In the current manuscript, the authors have conducted a pilot study using a targeted isotope dilution mass spectrometry assay and quantified Osteopontin in plasma of six metastatic triple negative breast cancer patients.

The article is well structured into sections and subsections. The article is eloquent, and English is professional. It is within the scope of the journal.

However, there are some comments that need to be addressed to improve the article. The detailed comments are below:

1) Abstract, line 35: OPN is a multifunctional protein and is important for normal physiological processes. The dysregulation in its levels (overexpression) promotes tumor progression. The sentence needs to be rephrased to improve clarity.

 We thank the reviewer for this comment and have revised the sentence as follows:

Original: These properties implicate OPN as a promotor of tumor progression.

Revised: OPN is involved in several normal physiological processes; however, OPN dysregulation leads to overexpression by tumor cells leading to immune evasion and increased metastasis.

2) Introduction: Is there a specific proteoform of OPN that is associated with the disease (cancer in this case)? For instance, elevated levels of glycosylated or phosphorylated forms? Not all proteoforms of OPN may be linked to cancer.

We agree that this is an important point. OPN has three splice variants (OPN-a,OPN-b & OPN-c) that are the focus of this manuscript. Other proteoforms result from the numerous phosphorylation (n=30) and glycosylation (n=5) sites. However, their role in cancer is unclear but we hope that improved assays will aid these efforts. We have modified lines 104-5 as follows and include reference to a comprehensive review of OPN in cancer.

Original: We evaluate several additional SIL peptides that correspond to the known OPN isoforms. 

Revised: We evaluated several additional SIL peptides that correspond to OPN splice variants OPN-a, OPN-b and OPN-c since their roles in cancer are the most characterised amongst the several known OPN isoforms.

3) Introduction, line 89: Authors have described studies that have attempted OPN quantification. It would be relevant to describe Macur et al (2019) (Reference 19), where the analytical workflow included the enrichment of peptides and LC-MRM-MS analysis with stable isotope standards (SIS) standards that enabled the OPN quantification upto 2.5 ng/mL. Authors can highlight the merits of the workflow used in the current study, however the detection limit of this assay is 39 ng/mL.

We thank the reviewer for this comment and have added more detail to the discussion in order to highlight the merits of the current workflow as suggested. We note that the methods for evaluating the detection limit differ; we used a protein spiked into plasma and Macur et al used a purified peptide. We have included similar data in order to allow for a direct comparison.

“Using the optimal conditions, the mass detection limit of the LC-MRM assay using synthetic peptides was 1.3, 1.5, 3.1 and 4.2 ng/ml for peptides QNLLAPQNAVSSEETNDFK, ANDESNEHSDVIDSQELSK, AIPVAQDLNAPSDWDSR, and YPDAVATWLNPDPSQK respectively. This sensitivity is similar to the sensitivity of 2.5 ng/ml reported by Macur et al.19”

4) Lines 217-224: For the pilot study, authors have used plasma from six triple negative breast cancer patients. However in line 221 and 222 authors mention patient 23. The sentence needs clarity. For instance, it can be mentioned as six patients namely or labelled 02, 03…and 23.

 We have added more detail as suggested by the reviewer.

 Original 

All plasma samples were collected from patients with recurrent metastatic disease and have previously undergone multiple lines of systemic chemotherapy.

 Revised

All plasma samples were collected from patients with recurrent metastatic disease and have previously undergone multiple lines of systemic chemotherapy. Sample were labelled with anonymized identifiers (02, 03, 05, 06, 09 and 23) and the blood collection date.

5) Lines 241-243: It is mentioned that plasma OPN levels increased with disease progression, then why plasma levels were lower in subsequent sampling? 

The reviewer is correct that OPN levels have been shown to increase between patients as the cancer stage increases from I to IV; however, very little data is available for OPN within the same patient at multiple time points. The studies that have been published show a complex behaviour; OPN increases with tumor size but also increases immediately after tumors are surgically removed but decreases due to chemotherapy. The text has been modified as follows to reflect this complexity.

Original

The correlation between plasma OPN levels and disease activity was examined in Patient 02 that had serial sampling between October, 2010 and April, 2011. Plasma OPN levels increased concomitantly with disease progression as observed by CT imaging; however, plasma OPN levels were lower in subsequent sampling despite the lack of change in disease activity based on CT imaging. OPN has many diverse biological roles associated with aggressive cell behaviour, tumor progression and metastasis (Bramwell et al, 2014). High levels of OPN are associated with an increased tumor burden and decreased survival (Wang et al, 2006). The differential expression of OPN in breast cancer is still poorly understood and confounding variables of the host response to systemic chemotherapy may play a role in levels of plasma OPN. 

Revised

The correlation between plasma OPN levels and disease activity was examined in Patient 02 that had serial sampling between October, 2010 and April, 2011. Where possible, plasma OPN was compared to tumor size assessed from CT imaging; however, since CT imaging is only performed every 2-4 months and blood draws are performed more frequently, this comparison is not possible at all time points. Plasma OPN levels increased concomitantly with disease progression as observed by CT imaging; however, plasma OPN levels were lower in subsequent sampling despite the lack of change in disease activity based on CT imaging. The specific disease progression observed for patient 02 was progression from primary to metastatic disease with new lesions being identified in the bone, which may have been facilitated by the increased OPN levels {Zuo, 2021 #30}. However, due to disease progression, the patient was also administered systemic chemotherapy, which has been shown to decrease OPN levels. 

6) Figure 1 Legend, line 264: The figure panels can be labeled and legend can be improved to aid the readers. Denote the short name used for the peptides either in Table1 or in the legend. 

Table 1 has been updated with a new column titled “ID” indicating each peptides short name.

7) Figures: All figures need improvement. The image resolutions are very low, and the axis labels are not legible in some cases. Proper scaling is required. Figure 2 – The axis label seems half cropped. Figure 4 – nothing is legible.

We apologize for the quality of the figures. They were all uploaded at high quality but we have no control over how they are processed by the journal submission system. However, we have included the images submitted below for your convenience.

Figure 1 – Analysis of OPN standard by LC-MRM-MS.

The assay consists of four peptides with four fragments measured for each peptide for a total of 14 MRMs. In this example, a standard solution at 200 ng/ml in buffer was analysed. The use of multiple fragments provides high specificity.

Figure 2 – Workflow and optimization of peptide recovery using addition extraction step

Diagrammatic depiction of OPN extraction procedure using single-tube extraction, digestion process (top panel). Recovery was initially low (panel A) when the manufacturer’s protocol was used but increased significantly when an additional extraction step was introduced (panel B).

Figure 3 – Quantification of OPN spike in normal plasma

OPN was spiked into control plasma at 400 ng/ml and processed using the optimised precipitation protocol. Quantification of the spiked protein (red traces) was performed using isotope dilution with a heavy labelled peptide (blue traces) for the four target peptides. The sum of four MRM transitions is shown.

Figure 4 – Quantification of OPN in plasma of metastatic breast cancer patients 

 Plasma OPN measurements based on the most sensitive peptide AIPVAQDLNAPSDWDSR from plasma of metastatic breast cancer patients. A) Extracted ion chromatogram (XIC) of a processed plasma sample (10 µL) following the established workflow. Plasma OPN concentration (red trace) was calculated using the ratio of the endogenous peak area over the heavy AQUA chain (blue trace). B) Bar plot representation of the OPN concentrations calculated from the plasma samples (mean + SD). The control sample is from pooled normal human plasma.

Figure 5: Serial OPN measurements and tumor size 

Serial tumor size measurement by CT reveals rapid tumor growth accompanied by spike in plasma OPN concentration in Patient 02 suggesting change in OPN concentrations may be used to monitor disease progression. 

Supplemental Figure S1 – Collision Energy Optimization curves. Optimization of collision energy for five target OPN tryptic peptides. A 200 ng of tryptic digest of OPN protein was injected for parameter optimization. 

Reviewer #2: Osteopontin was proposed as a potential biomarker in different diseases including breast cancer. Yet, to verify its biomarker potential in certain disease osteopontin needs to be verified in larger groups of clinical samples representative for a certain disease or its state. This in turn requires an efficient, high throughput and reliable methodology for sample processing and measurement. This goal was achieved by the authors of the manuscript, who developed a LC-MRM-MS-based assay with SIL standards for quantification of OPN in plasma with an efficient sample preparation method that included protein precipitation, digestion and clean-up on a ProTrap XG filter cartridge. They verified the assay on a set of breast cancer patients' plasma. In my opinion the manuscript is good, although needs some minor corrections/comments to be published in PLOS ONE:

(1) Please provide the ProteomeXchange number for your data-now it is missing. 

All raw data has been uploaded as submission PASS04822 to PeptideAtlas.

(2) As I understood, the assay is intended to detect all OPN isoforms present in the sample, without distinguishing between them. Is any of the peptides unique for a particular isoform? Moreover, the concentration calculation is based on the signal detected for all the investigated peptides. Then, please comment on the impact of presence/absence of certain isoform on the final quantification results. The authors also mention post-translational modifications of OPN. PTMs were also detected on OPN peptides that they choose for quantification (please see eg. UniProtKB), which will make such modified peptide not detected by the developed LC-MRM-MS assay. Please comment on impact of presence/absence of the PTMs on the final quantification results and possible method limitations. 

The reviewer is correct; the assay is intended to distinguish between isoform since the peptides chosen span deletions present in OPN-b (deletion of exon 5) and OPN-c (deletion of exon 4). OPN-a is the full length and would have signal for all four peptides. The following text was added to better explain this mapping:

As expected, this full length OPN-a isoform has all four peptides present. As isoform OPN-b lacks amino acids 59-72 (exon 5 deletion), no signal for peptide QNLLAPQNAVSSEETNDFK would be observed for OPN-b. In a similar fashion, OPN-c lacks amino acids 31-57 (exon 4 deletion) and no signal for peptide YPDAVATWLNPDPSQK would be observed for OPN-c.

(3) Are LODs, LOQs and concentrations in patients's samples presented as mean +/- SD? Please specify and comment on the obtained results in the manuscript (line 180-187). 

Original:

Background values plus 3 times the standard deviation or five time the standard deviation were used to calculate the LOD and LOQ, respectively.

Revised:

LOD’s for each peptide were determined by first averaging the response of quantifier transitions for 8 replicate blank injections. The blank response was defined as the average of the replicate blanks plus 3x the standard deviation of the blank response areas. Single point calibration between the calculated blank response and ILIS peptides spiked at 2 concentration levels were averaged to determine per peptide LOD values. 

(4) Were calibration curves prepared for the investigated peptides or was it a single point calibration? Please specify clearly in the manuscript. If available, please add information about linear range of the assay. 

Revised:

Sample peptide concentration values were determined from a single point calibration comparison of the average of duplicate quantifier transitions with ILIS spiked peptides (18 fmol/uL each). Resulting values were converted to ng/mL units using the molecular weight of OSTP (36229.53 g/mol).

(5) What software was used for data acquisition, extraction and processing? What parameters of the software were applied for data processing? 

Revised:

 All mass spectrometry data was acquired using Analyst version 1.6.2 with peak area integration using Skyline version 22.2.0.255. All data analysis utilized in-house generated scripts written in R (version 4.2.1). 

(6)I think it should be 'transtuzumab' instead of 'tranztuzumab' (line 55-56). 

 The reviewer is correct; “traztuzumab” has been corrected to “trastuzumab”.

(7) It would be informative to give information about patients, where the cancer type and stage would be assesed by one of the clinically applied scales. Please include this information if available. 

A Table summarizing available information on the patients has been added to the Supplementary section.

Patient ID Type of BC Histological finding Metastatic sites Lines of therapy

2 TN IDC 3 1

3 TN NA NA NA

5 TN NA NA NA

6 TN NA NA NA

9 TN IDC 2 1

23 TN IDC 1 1

IDC: Invasive ductal carcinoma; TN: Triple negative; NA: not available

Reviewer #3: In their manuscript “A targeted isotope dilution mass spectrometry assay for Osteopontin quantification in plasma of metastatic breast cancer patients”, Andrew Leslie et al. developed 4 SID-SRM LC-MS assays for the quantification of the most relevant Osteopontin proteoforms. They also demonstrate the ability to quantify OPN in plasma samples using a slightly optimized sample preparation workflow that does not include any OPN enrichment step, which would be required to achieve sufficient sensitivities. As such, the targeted MS assays are sound, but lack sensitivity in real plasma samples. Nonetheless, the assays are a good starting point to use targeted MS for OPN quantification and are certainly interesting for researchers working in the field. I summarized my comments below that need to be addressed before publication.

Comments:

Line 212: The authors claim to recover 100% of the 4 selected peptides with their optimized sample preparation workflow. It is not clear from Figure two how this was determined. This is not too easy. Ideally, a reference protein would be spiked before the sample preparation in know amounts and is then determined by using heavy reference peptides spiked directly before LC-MS analysis. The ratio will then reflex the losses taking place during sample preparation. The authors should explain in detail how the 100% were determined or tune down this sentence saying that recovery of hydrophobic peptides was considerably improved by the additional elution step. 

Original:

When analysing recombinant OPN spiked into normal plasma, the additional elution step returned the recovery to 100% for all but one peptide, ANDESNEHSDVIDSQELSK, which was the most hydrophilic of the four peptides (Figure 2).

Revised:

When analysing recombinant OPN spiked into normal plasma, the additional elution step significantly improved peak areas for all but one peptide, ANDESNEHSDVIDSQELSK, which was the most hydrophilic of the four peptides (Figure 2).

Line 225: The OPN concentrations found here ranged between 17 and 53 ng/ml. However, above, the authors describe the LOD/LOQs of the single peptide assays developed to be much higher. Only LOD of one peptide was at 39 ng/ml, LOQ at 97 ng/ml. Does this mean that the OPN concentrations determined are all below LOQ? How useful are the assays actually with LOQs being much above the actual OPN concentrations found in most plasma samples? This and further improvement possibilities to boost sensitivity should be discussed! 

The assay is more simple and more sensitive that a previous publication in POLS One using immune-MS for OPN analysis in tissue. However, in spite of this, the reviewer is correct that further improvement in the LOQ are needed in order to allow for the assay to be useful clinically for the analysis of plasma OPN isoforms in cancer patients. 

The following has been added to the results section (line 237):

These values are between the assay LOD and LOQ, therefore, carry a large uncertainty and point to the need for further improvement in assay sensitivity.

Also, the conclusions have been revised:

Original:

The sensitivity of the method, with a detection limit of 39 ng/mL OPN in plasma, is sufficient for detection of OPN in patients with elevated OPN levels, such as those with high tumor burden. However, further increases in sensitivity are needed for use of this assay to measure low ng/mL levels of OPN and to provide full coverage of the various isoforms of OPN. 

Revised:

The sensitivity of the method, with a detection limit of 39 ng/mL OPN in plasma, can detect OPN levels in metastatic breast cancer patients with high tumor burden but is insufficient for detection of OPN in early-stage breast cancer patients. The sensitivity is likely sufficient for the analysis of tumor biopsy material since OPN levels are often highly elevated at the tumor site. However, further increases in sensitivity are needed to enable measurement of OPN isoforms at single ng/ml levels for it to be useful in a liquid biopsy. This is advantageous as liquid biopsies are less invasive and can be performed at multiple time points to enable studies on correlation between OPN levels and disease progression or treatment response. In addition, further increases in assay sensitivity would enable full coverage of the various isoforms of OPN in plasma in order to better understand the role of OPN isoforms in cancer progression. 

The entire data analysis part including a statical analysis is missing in the manuscript and should be added. 

Please see Reviewer 2 Comments 3,4 above for inline addition of statistical analysis.

Figure 2: It would be helpful to add the time span between the spins into the schematic overview to better grasp the workflow. In particular, when adding trypsin, some incubation time (overnight) is required before spinning down the peptides. 

Revised:

Figure 2:

Add “(Overnight at 37 ºC)” below ‘5 µg trypsin’ vial label.

Add “(5 min)” below each Spin arrow

Figure 5: Why was there no Tumor size measurement at 2011-01-10? 

Tumor size measurement is typically performed every 2-4 months as more frequent intervals rarely show sufficient changes to inform or effect clinical decision-making. As a CT scan had been performed on 2010-12-20, a scan was not indicated on 2011-01-10. We have added the following text to provide this context:

Where possible, plasma OPN was compared to tumor size assessed from CT imaging; however, since CT imaging is only performed every 2-4 months and blood draws are performed more frequently, this comparison is not possible at all time points.

The raw MS data needs to be public available. Please upload to a public repository, like PANORAMA. 

Please see Reviewer #2, Comment #1

6. PLOS authors have the option to publish the peer review history of their article (what does this mean?). If published, this will include your full peer review and any attached files.

Do you want your identity to be public for this peer review? For information about this choice, including consent withdrawal, please see our Privacy Policy.

Reviewer #1: Yes: Ankita Punetha

Reviewer #2: No

Reviewer #3: No

In compliance with data protection regulations, you may request that we remove your personal registration details at any time. (Remove my information/details). Please contact the p

---

## [Decision Letter · Decision Letter 1]

18 May 2023

PONE-D-23-02025R1A targeted isotope dilution mass spectrometry assay for Osteopontin quantification in plasma of metastatic breast cancer patientsPLOS ONE

Dear Dr. Pinto,

Thank you for submitting your manuscript to PLOS ONE. After careful consideration, we feel that it has merit but does not fully meet PLOS ONE’s publication criteria as it currently stands. Therefore, we invite you to submit a revised version of the manuscript that addresses the points raised during the review process.

ACADEMIC EDITOR: Would you please go through the comments raised by reviewer # 2 and amend the MS accordingly. Afterward, proofread the text for grammar and syntax errors, if any.

We look forward to receiving your revised manuscript.

Kind regards,

A. M. Abd El-Aty

Academic Editor

PLOS ONE

Journal Requirements:

Reviewers' comments:

Reviewer's Responses to Questions

**Comments to the Author**

1. If the authors have adequately addressed your comments raised in a previous round of review and you feel that this manuscript is now acceptable for publication, you may indicate that here to bypass the “Comments to the Author” section, enter your conflict of interest statement in the “Confidential to Editor” section, and submit your "Accept" recommendation.

Reviewer #1: All comments have been addressed

Reviewer #2: All comments have been addressed

Reviewer #3: (No Response)

2. Is the manuscript technically sound, and do the data support the conclusions?

Reviewer #1: Yes

Reviewer #2: Yes

Reviewer #3: Yes

3. Has the statistical analysis been performed appropriately and rigorously? 

Reviewer #1: Yes

Reviewer #2: I Don't Know

Reviewer #3: Yes

4. Have the authors made all data underlying the findings in their manuscript fully available?

Reviewer #1: Yes

Reviewer #2: Yes

Reviewer #3: Yes

5. Is the manuscript presented in an intelligible fashion and written in standard English?

Reviewer #1: Yes

Reviewer #2: Yes

Reviewer #3: Yes

6. Review Comments to the Author

Reviewer #1: The authors have diligently addressed all the raised concerns. The manuscript after revision has significantly improved.

In the Supplemental Figure S1 – Collision Energy Optimization curves, the axes labels are not legible. It is advised to increase the font size for better readability.

Reviewer #2: The authors developed an OPN quantifcation methodology that does not require use of antibodies and with a simple sample preparation procedure with LOD at 39 ng/mL, which is a good result. However, the assay has got insufficient sensitivity to quantify OPN in the samples from some breast cancer patients, as the detected concentrations were between 17 and 53 ng/mL. Therefore, I think it would be fair to the potential reader to state also in the abstract: (1) OPN levels detected in the pilot study plasma samles, and, (2) that the assay is useful rather for plasma from patients of high grade tumors, where the expected OPN concentration is higher.

Reviewer #3: The manuscript has been considerably improved by addressing all comments satisfactorily. It will be very useful to readers and is now ready for publication.

7. PLOS authors have the option to publish the peer review history of their article (what does this mean?). If published, this will include your full peer review and any attached files.

Reviewer #1: No

Reviewer #2: No

Reviewer #3: **Yes: **Alexander Schmidt

---

## [Author Response · Author response to Decision Letter 1]

6 Jun 2023

Reviewer #1: The authors have diligently addressed all the raised concerns. The manuscript after revision has significantly improved.

In the Supplemental Figure S1 – Collision Energy Optimization curves, the axes labels are not legible. It is advised to increase the font size for better readability.

 An improved version of this figure with larger fonts is included

Reviewer #2: The authors developed an OPN quantifcation methodology that does not require use of antibodies and with a simple sample preparation procedure with LOD at 39 ng/mL, which is a good result. However, the assay has got insufficient sensitivity to quantify OPN in the samples from some breast cancer patients, as the detected concentrations were between 17 and 53 ng/mL. Therefore, I think it would be fair to the potential reader to state also in the abstract: (1) OPN levels detected in the pilot study plasma samles, and, (2) that the assay is useful rather for plasma from patients of high grade tumors, where the expected OPN concentration is higher.

 The following was added to the abstract:

The assay was applied to the analysis of plasma OPN in metastatic breast cancer patients, where levels from 17 to 53 ng/mL were detected. The sensitivity of the method is higher than previously published methods and sufficient for OPN detection in large, high grade tumors but still requires improvement in sensitivity to be widely applicable.

---

## [Decision Letter · Decision Letter 2]

15 Jun 2023

A targeted isotope dilution mass spectrometry assay for Osteopontin quantification in plasma of metastatic breast cancer patients

PONE-D-23-02025R2

Dear Dr. Pinto,

We’re pleased to inform you that your manuscript has been judged scientifically suitable for publication and will be formally accepted for publication once it meets all outstanding technical requirements.

Kind regards,

A. M. Abd El-Aty

Academic Editor

PLOS ONE

Additional Editor Comments (optional):

Reviewers' comments:

Reviewer's Responses to Questions

**Comments to the Author**

1. If the authors have adequately addressed your comments raised in a previous round of review and you feel that this manuscript is now acceptable for publication, you may indicate that here to bypass the “Comments to the Author” section, enter your conflict of interest statement in the “Confidential to Editor” section, and submit your "Accept" recommendation.

Reviewer #1: All comments have been addressed

Reviewer #2: All comments have been addressed

2. Is the manuscript technically sound, and do the data support the conclusions?

Reviewer #1: (No Response)

Reviewer #2: Yes

3. Has the statistical analysis been performed appropriately and rigorously? 

Reviewer #1: (No Response)

Reviewer #2: I Don't Know

4. Have the authors made all data underlying the findings in their manuscript fully available?

Reviewer #1: (No Response)

Reviewer #2: Yes

5. Is the manuscript presented in an intelligible fashion and written in standard English?

Reviewer #1: (No Response)

Reviewer #2: Yes

6. Review Comments to the Author

Reviewer #1: (No Response)

Reviewer #2: I have not got any more comments to the Authors. All the prevoius comments have already been adressed by the Authors in this revieved verisoin of the manuscript.

7. PLOS authors have the option to publish the peer review history of their article (what does this mean?). If published, this will include your full peer review and any attached files.

Reviewer #1: No

Reviewer #2: No

---

## [Editor Report · Acceptance letter]

19 Jun 2023

PONE-D-23-02025R2 

A targeted isotope dilution mass spectrometry assay for Osteopontin quantification in plasma of metastatic breast cancer patients 

Dear Dr. Pinto:

I'm pleased to inform you that your manuscript has been deemed suitable for publication in PLOS ONE. Congratulations! Your manuscript is now with our production department. 

Kind regards, 

on behalf of

Prof. A. M. Abd El-Aty 

Academic Editor

PLOS ONE